# The Swing of Lipids at Peroxisomes and Endolysosomes in T Cell Activation

**DOI:** 10.3390/ijms21082859

**Published:** 2020-04-19

**Authors:** Sara G. Dosil, Amelia Rojas-Gomez, Francisco Sánchez-Madrid, Noa B. Martín-Cófreces

**Affiliations:** 1Servicio de Inmunología, Hospital Universitario de la Princesa, Universidad Autónoma de Madrid (UAM), Instituto de Investigación Sanitaria Princesa (IIS-IP), 28006 Madrid, Spain; dosilsara@gmail.com (S.G.D.); rojasgomezamelia@gmail.com (A.R.-G.); 2Centro Nacional de Investigaciones Cardiovasculares (CNIC), 28029 Madrid, Spain; 3Centro de Investigación Biomédica en Red de Enfermedades Cardiovasculares (CIBERCV), 28029 Madrid, Spain

**Keywords:** immune synapse, lipid mediator, mitochondria, peroxisome, metabolomics

## Abstract

The immune synapse (IS) is a well-known intercellular communication platform, organized at the interphase between the antigen presenting cell (APC) and the T cell. After T cell receptor (TCR) stimulation, signaling from plasma membrane proteins and lipids is amplified by molecules and downstream pathways for full synapse formation and maintenance. This secondary signaling event relies on intracellular reorganization at the IS, involving the cytoskeleton and components of the secretory/recycling machinery, such as the Golgi apparatus and the endolysosomal system (ELS). T cell activation triggers a metabolic reprogramming that involves the synthesis of lipids, which act as signaling mediators, and an increase of mitochondrial activity. Then, this mitochondrial activity results in elevated reactive oxygen species (ROS) production that may lead to cytotoxicity. The regulation of ROS levels requires the concerted action of mitochondria and peroxisomes. In this review, we analyze this reprogramming and the signaling implications of endolysosomal, mitochondrial, peroxisomal, and lipidic systems in T cell activation.

## 1. Introduction

Cellular adaptive immunity requires the interaction of antigen presenting cells (APC) with T cells bearing T cell receptors (TCR), which need to be specific for the antigen: major histocompatibility complex (MHC) combination presented by the APC. The intimate contact between the APC and the T cell has come to be known as the immune synapse (IS) [1,2]. The IS is a dynamic, highly organized macromolecular structure initiated by antigen recognition through the TCR. It is worth mentioning that this structure is also formed in other immune events, i.e., between natural killer and infected/tumor cells allowing their cytolytic activity or between macrophages or mast and T cells triggering the inflammatory response. Among other functions, the IS promotes the bidirectional communication between the T cell and the APC [1,3,4,5]. During antigen presentation, the T lymphocyte relocates receptors, cytoskeletal components, and specific organelles to the contact site with the APC. These drive actin polymerization and reorganization, forming a contractile actomyosin ring that shapes the IS [6,7]. These signals also drive de novo polymerization of microtubules (MTs) from the centrosome, which is polarized towards the T cell:APC interface [8]. Once this reorganization is achieved, the cytoskeleton guides the redistribution of the endolysosomal system (ELS) towards the IS, facilitating the subsequent polarized secretion and endocytic recycling of receptors at the T cell-APC contact site [9,10]. Mitochondria also relocate to the IS, where they play a role in the maintenance of Ca^2+^ fluxes along the T cell activation cycle [11,12,13,14].

To establish a correct communication between the IS-engaged cells, the information travels back and forth between the APC and the T cell. Much attention has been devoted to the activating effects of the APC on the T cell [15]. On the other hand, the engaged T cell also influences APC function using soluble factors (e.g., cytokines and chemokines), exosomes, and other types of secretory vesicles that are released during the APC-T cell contact [16,17,18,19,20]. This crosstalk favors the functional modulation of the APC and T cells that form the IS and, eventually, the optimal priming, activation, and T cell responses against cognate antigens.

Upon IS formation, mitochondria and peroxisomes are major organelles that mediate lipid metabolism changes and, in turn, T cell activation signaling pathways. Mitochondria polarize to the immune synapse and act as major regulators of lipid metabolism, reactive oxygen species (ROS) production and calcium fluxes. In addition, peroxisomes act as collaborators of mitochondria in the regulation of both lipid and ROS levels. The ELS is a dynamic cell compartment with several organelles and the plasma membrane that mediates vesicle and exosome delivery between the APC and the T cell. It comprises two different compartments: early and late endosomes. Late endosomes form intraluminal vesicles (ILVs) by the invagination of their limiting membrane [21], becoming the so-called multivesicular bodies (MVB). These ILV-enriched organelles are directly involved in sorting and recycling of proteins from the plasma membrane to intracellular compartments and vice versa [22]. Furthermore, they are directly related to exosome formation and their sorting towards the APC [23,24,25]. As a result of these transport events the plasma membrane has a heterogeneous profile composed of co-existing domains enriched in different proteins and lipids. These may serve to regulate cell-to-cell communication and activation [26]. T cell activation, in addition, is accompanied by a metabolic reprogramming. This process requires high ATP consumption that results in an increase of lipid and ROS levels. These changes in the T cell modulate plasma membrane composition, protein signaling pathways and cytoskeleton organization. These effects are accompanied by changes in metabolic processes that, in turn, regulate ROS levels. In this review we offer to the reader an overview of the changes in intracellular organelles and lipid metabolism associated to T cell activation through the IS.

## 2. Endolysosomal System and Multivesicular Bodies

The endolysosomal system is the main regulator of metabolic homeostasis. It allows the recycling of previously synthesized molecules. Endosomes are identified by the presence of Rab GTPase proteins. The ELS uses MVB morphogenetic pathways to reach the lumen of lysosomes, where intracellular and extracellular cargoes are degraded. This process acts in concert with autophagy, in which damaged organelles and aggregated proteins are recycled through lysosomal pathways. Autophagy is induced in IS-forming dendritic cells through ATG16L; it regulates mTORC activity, acting as a negative feedback modulator by destabilizing the synapse. Specific mutations of this protein have been described to lead to hyperstable T cell:APC interactions and possibly contributing to T cell hyperactivity in Crohn’s disease patients [27]. On the T cell side, the intraflagellar transport system (IFT) regulates the biogenesis of lysosomes [28] and the retrograde transport towards intracellular endosomes upon immune synapse formation, favoring CD3 and LAT recruitment and recycling [29]. In addition, there are other non-vesicular and vesicular mechanisms that target selective cargoes to lysosomes, such as chaperone-mediated autophagy and microautophagy [30,31].

Recycling and delivery of molecules at the IS interface is important for a balanced downstream signaling and full T cell activation. For example, both the TCR/CD3 complex and the downstream scaffold protein LAT (Linker for Activation of T Cells) are subjected to a tight regulation along this process [16,17]. Vesicle traffic constitutes a basic component of this machinery, and cytoskeletal components modulate its dynamics [18,19,20]. MVB relocate to the IS and help the establishment of the actin-rich ring through clathrin and associated components, such as Hrs [32]. Hrs (Hepatocyte growth factor-regulated tyrosine kinase substrate) is an endosome-associated regulator for vesicular transport and protein sorting [33].

This process enables the integrin-based adhesive part of the IS (Peripheral Supramolecular Activation Clusters, pSMAC) to seal the intercellular space between the T cell and the APC [1,8]. LBPA (lysobisphosphatidic acid) represents about 15 % of total phospholipids in MVB and is only detected at the IS, which strongly argues for a localized synthesis from a phospholipidic precursor at this location [34]. LBPA appears late in the IS, specifically after MVB fusion [11], which provides the plasma membrane with specific lipids that may regulate other processes. The fusion of MVB at the IS allows the release of their content, including ILVs as exosomes. Exosome membranes are enriched in cholesterol, sphingolipids such as sphingomyelin (SM) and GPI-anchored proteins, as well as tetraspanins and LAMPs (lysosome-associated membrane proteins) [35,36,37,38]. The delivery of exosomes from MVB at the IS interface allows their capture by the APC, which causes changes in the recipient cell [25,39,40,41,42].

Extracellular vesicle biogenesis follows two main routes. One involves the formation of endosomal patches found at the plasma membrane and the fusion of endosome membranes with the plasma membrane [43]. The second is slower because it requires the invagination of endosomal membranes to form the ILVs and their subsequent release through exocytosis. Protein sorting to ILVs is mediated by the ESCRT (endosomal sorting complexes required for transport) machinery after its interaction with K63-ubiquitinated proteins [44,45]. Different components of the ESCRT machinery, such as ESCRT-0 and ESCRT-1 (TSG101), contribute to MHC-II sorting to exosomes and biogenesis of membrane-derived exosomes [46]. On the other hand, ESCRT-III seems to play a negative role, and its inactivation increases ILV production [47]. The cell content of MVB increases with T cell activation. This increase is regulated by diacylglicerol kinase α (DGKα) through a dual role. On one hand, DGKα produces phosphatidic acid from diacylglicerol (DAG), which negatively regulates T cell activation; depletion of this enzyme prevents CD28-dependent T cell anergy [48]. On the other hand, DGKα translocates to MVB upon TCR activation and its inhibition increases mature MVB and exosome secretion [49].

## 3. Mitochondria and Peroxisomes: Modulators of the Immune Synapse

During differentiation, T cells increase their pool of mitochondria to pay the energetic cost of this process. In addition, these organelles are also involved in lipid synthesis, calcium homeostasis, apoptosis, signaling, and cell progression [50,51]. These processes place mitochondria as important organelles during immune synapse formation and T cell activation. After antigen encounter and receptor stimulation, mitochondria are polarized to the IS [14,52,53]. This is related to the regulation of calcium fluxes and to the production of the energy required for activation. As we describe below, mitochondrial metabolism is critical for T cell activation through production of mitochondrial ROS (mROS) [54,55,56].

Peroxisomes, in collaboration with mitochondria, prevent cell toxicity through ROS elimination. Besides oxidative modulators, these organelles are sites for Fatty Acid β-Oxidation (FAO). They also promote D-amino acid degradation, polyamine oxidation, and synthesis of various cholesterol precursors and plasmalogens [57]. Peroxisome proliferation and degradation are regulated processes and, in contrast to mitochondria, these organelles are unable to fuse with each other. There are two types of peroxisome biogenesis processes: growth and division of pre-existing organelles [58], and de novo formation from the endoplasmic reticulum (ER) [59]. Degradation of these organelles occurs by micro- and macro-autophagy or by 15-lipoxygenase-mediated autolysis [60]. The association between peroxisomes and T cell maturation has remained elusive, although some evidence links PPAR (Peroxisome Proliferator-Activated nuclear Receptor) to T cell development. PPARs are a family of ligand-activated transcription factors that, when dysregulated, affect a variety of physiological processes (lipid metabolism, cellular differentiation, and cancer). For example, alteration of cholesterol metabolism can affect TCR signaling or FAO [61]. In addition, PPARβ overexpression is related to inhibition of peripheral αβ T cell proliferation, while the γδ T cell population remains unaffected [62]. Thus, mitochondria and peroxisomes are key players in T cell activation. Calcium fluxes, ATP production, and ROS levels are processes directly modulated by these organelles, as illustrated in Figure 1.

## 4. Reactive Oxygen Species as Modulators of T Cell Activation

ROS are chemically reactive free radicals with one unpaired electron in their outer orbit. The most familiar ones are superoxide, hydrogen peroxide, hydroxyl radical, and singlet oxygen. These free radicals are generated as the result of a partial reduction of oxygen, a process that occurs mainly in mitochondria and peroxisomes. This output is directly related with the oxidative and fatty acid metabolism and there is an active interplay between both organelles [63,64]. In fact, they follow common steps in their course, such as dehydrogenation, hydration, and thiolytic cleavage for FAO [65]. In addition, mitochondria and peroxisomes have an equilibrium between oxidant and antioxidant agents to avoid an oxidative stress that could ultimately alter their function. A misregulation of this balance could induce an uncontrolled proliferation, eventually leading to cancer [57,63,64,66,67,68].

ROS have always been related to harmful by-products that cause DNA damage, genomic instability, or protein dysfunction, leading to cell senescence or cell death caused by apoptosis [69,70,71]. However, in the last few years this view has changed. Variations in the oxidative components during different cell processes are able to regulate cell metabolism. In this sense, low to moderate ROS levels contribute to signaling pathways implicated in cell growth, death, and migration [72]. For example, phosphatases, kinases, and transcription factors can be modulated by oxidative transitions. In addition, ROS-related modifications can alter the function of some proteins by modulating their associated signaling pathways [73,74]. T cells are not an exception and variations in ROS levels have a role during their activation and differentiation stages. The main ATP sources of circulating naive T cells (T_n_) are oxidative phosphorylation (OXPHOS) and FAO [75,76]. T cell activation causes an abrupt increment of ATP consumption leading to metabolic reprogramming. The catabolic state gives rise to an anabolic metabolism that provides the energy necessary to proliferate, differentiate and become effector T cells (T_eff_). This is characterized by increased nutrient uptake causing, in consequence, a gain in glycolysis products. This shift activates signaling pathways, transcription factors and effector molecules [77,78,79,80,81]. Additionally, TCR ligation increases the production of ROS by OXPHOS and NADPH oxidases (NOXs), a family of plasma membrane-associated oxidases that mediate proliferation and differentiation [55,82,83]. This is complemented with the crucial role of mROS, which activate members of the NFAT family of transcription factors and Myc [54]. Additionally, they also activate signaling molecules such as mTOR, NF-kB, and AP-1. These effects directly relate mROS with an IL-2 dependent proliferative status and with a metabolic reprogramming of T effector cells [55,56]. After antigen clearance, only a small proportion of T cells survive, becoming memory T cells (T_m_). These cells remain primed for a future encounter with the same antigen [84,85]. T_m_ do not have the same energy demand as T_eff_, losing those glycolytic requirements and recovering the metabolic profile of T_n_. To avoid apoptosis and cell death, this is accompanied by an increment in mitochondrial FAO and the number of mitochondria [76]. In addition, low amounts of metabolic ROS promote increased lifespan and response capacity of these T_m_ [86].

## 5. Lipids as a Source of Energy in Metabolic Reprogramming after T Cell Activation

As mentioned above, when T cells are activated, they undergo metabolic reprogramming, acquiring a biosynthetic status. There is a switch from pyruvate oxidation via the tri-carboxylic acid cycle and FAO to aerobic glycolysis, together with induction of the pentose-phosphate pathway, glutaminolysis and de novo fatty acids synthesis (FAS). This is controlled by changes in the metabolic transcriptome and the subsequent induction of the transcription factors Myc and HIF1α [54]. The metabolites resulting from the induction of those pathways are used as the nitrogen and carbon sources necessary for a range of biosynthetic precursors of lipids and polyamines. The reprogramming of activated T cells includes also a marked downregulation of mitochondria-dependent FAO, with a decrease in the levels of products such as carnitine [54]. Another regulator playing an important role in de novo FAS induction is the mTOR1 complex (mTORC1) [87]. Some studies demonstrate that the diminished function of mTORC1 due to the lack of its scaffolding protein Raptor causes an impairment in de novo TCR induced lipid biosynthesis [88]. MTORC1 acts in conjunction with Sterol Regulatory Element Binding Proteins (SREBPs), which are a group of transcription factors acting as essential regulators of cholesterol synthesis. Thus, the mTORC-SREBP pathway activates the lipid biosynthetic program [88].

Although T cells obtain their energy preferably from glucose, the oxidation of fatty acids is also an important source of ATP. FAO occurs in the mitochondrial matrix and requires the entry of the fatty acid attached to carnitine through the mitochondrial membrane with the contribution of carnitine-palmitoyl-CoA transferase-1 (CPT1) [89,90]. Fatty acid entry into mitochondria is regulated by a feedback system: when acetyl-CoA carboxylase is inactivated by an AMPK-induced phosphorylation, the concentration of its product malonyl-CoA decreases, facilitating the activity of CPT1 [91,92]. This process is especially relevant for the development and maintenance of regulatory T cells (T_reg_) and Tissue-Resident Memory T cells (TRM) [93,94].

## 6. Lipid second Messengers at the Immune Synapse (IS)

Phosphoinositide molecules have a key function in TCR downstream signaling. Phosphatidylinositol-4,5-bisphosphate (PIP_2_) plays structural and regulatory roles at the IS, both directly and indirectly through its derivatives. As a result of TCR/CD3 triggering, LAT recruits phospholipase C (PLC) and contributes to its activation. PLC converts PIP_2_ into inositol-1, 4, 5-trisphosphate (IP_3_) and diacylglicerol (DAG). PIP_2_ production at the IS is necessary to replenish the plasma membrane stores, since it represents less than 1% of membrane lipids. Sustained PIP_2_ turnover at the IS depends on CD28 co-stimulation in a PI5PKI-dependent manner and prevents T cell anergy [95,96,97]. PIP_2_ activates phospholipase D (PLD), which produces phosphatidic acid and activates PI5PKI, creating a positive feedback loop [98]. PIP_2_ is also a regulator of actin cytoskeleton assembly and the processes of endocytosis and exocytosis. ERM proteins (ezrin, radixin, and moesin) are found in vesicles. They adopt an open/active conformation upon binding to PIP_2_, facilitating the binding of a plethora of plasma membrane proteins—including adhesion and signaling receptors and tetraspanin scaffold proteins—to the actin cytoskeleton [99,100]. IP_3_ allows a transient opening of ER Ca^2+^ channels. The release of this secondary messenger to the cytosol regulates cell signaling, cytoskeletal reorganization, vesicular traffic, and secretion [8]. In contrast, DAG remains in the plasma membrane, preferentially accumulated at the cSMAC (Central Supramolecular Activation Clusters). Additionally, it is involved in the reorganization of the tubulin cytoskeleton through the control of centrosome polarity [101]. Finally, the joint action of DAG and the released Ca^2+^, allows the recruitment of protein kinase C (PKC) to the plasma membrane, its activation and, in turn, the beginning of signaling cascades [102].

Fatty acids also play a regulatory role in T cell activation as post-translational labels. For example, palmitoylation or myristoylation of several proteins of TCR downstream signaling leads to their recruitment to the IS. This enables the propagation of the signal inside the cell [103,104,105]. Furthermore, prenylation consists in the covalent attachment of a farnesyl or a geranylgeranyl isoprenoid group, which allows higher hydrophobicity and interaction with target membranes. Small GTPases including Ras, Rac, Rab, and Rho—with roles in signal transduction, cytoskeletal regulation and intracellular vesicle trafficking—undergo prenylation [106]. Acetylation is another post-translational modification important in signaling in both resting and activated T cells [107]. It derives from the key lipid metabolite acetyl-CoA. Acetyl groups modify histones and trigger epigenetic mechanisms which can alter the communication of the T cell with other immune cells, by regulating cytokine gene expression [108]. These changes are related to several chronic diseases [109].

## 7. Bioactive Sphingolipids and T Cell Activation

Sphingolipids (SL) are one of the most abundant components of the plasma membrane [110]. Bioactive sphingolipids play a role in the regulation of TCR signal transduction and protein sorting during T cell activation. [111]. This type of SL forms membrane domains together with sterols. These domains, together with compartmentalized membrane proteins and their proximal membrane associated signaling components, constitute the so-called lipid microdomains or lipid rafts [112]. The group of bioactive sphingolipids is composed of ceramide (Cer), sphingosine (Sph), ceramide-1-phosphate (C1P), and sphingosine-1-phosphate (S1P) [111]. The dynamics of bioactive sphingolipids and their implication in T cell activation are depicted in Figure 2.

Ceramide has a central function in the metabolism of bioactive sphingolipids; it is the substrate of a variety of enzymes that convert this lipid in the rest of bioactive SLs or other important lipids of the SL metabolism, as SM or glycosphingolipids [111,113]. SM is a key player in T cell activation and an important component of both the plasma membrane and lipid droplets [114]. SM is hydrolyzed at the membrane of diverse cellular components such as lysosomes by the corresponding isoform of the enzyme sphingomyelinase (neutral (nSMase) or acid (aSMase)). This reaction generates the products phosphocholine and Cer [113]. Then, the subsequent release of Cer within the T cell membrane leads to the formation of ceramide-enriched membrane microdomains at the IS [112]. Due to their biophysical properties, these structures compartmentalize receptors and their proximal signaling proteins, regulating TCR signaling. The maintenance of TCR signaling through CD3ζ and ZAP70 polarization to the IS requires nSM2. Furthermore, nSMase plays a role in the dynamics of polarization and stabilization of the microtubular system and the microtubule-organizing center. This is mediated by the recruitment of PKCζ and its downstream substrate Cdc42, which acts as organizer of cell polarity during IS maintenance [114]. Cer released from SM by SMases is used to synthetize de novo the rest of bioactive SL at different regions and compartments. It can also be recirculated to the cell membrane via endosomal pathway [111]. In relation to vesicular transport, there is experimental evidence supporting that nSMase2 triggers the budding of exosomes into MVB in T cells [39,115].

S1P and C1P are formed by the enzymatic action of kinases, which phosphorylate sphingosine and ceramide, respectively. S1P regulates lymphocyte egress into circulation through interaction with its receptor S1PR. S1PR association with CD69 [116,117] downregulates its membrane expression, allowing prolonged lymphocyte retention in inflamed tissues [118]. Due to its involvement in cell trafficking, S1P mediates cancer cell growth, proliferation, survival, and is related to inflammatory and autoimmune diseases such as asthma, atherosclerosis, Crohn’s disease, diabetes, and osteoporosis [119,120,121,122]. In addition, this bioactive SL is able to bind to transcription factors as HDAC1/2 or PPARɣ and modulates T cell metabolism to help differentiation, changing T cell capability of anti-tumor response [119]. C1P, in turn, increases the intracellular Ca^2+^ concentration in TCR-activated cells, as ceramide does. In contrast to Cer, C1P acts by promoting IP_3_ production, thereby inducing Ca^2+^ release from the ER and the opening of store-operated calcium channels (SOCC) at the plasma membrane [123]. Furthermore, C1P is tightly related with eicosanoid biosynthesis during immune response, activating initial rate-limiting enzymes in eicosanoid biosynthesis and inducing arachidonic acid release in the early stages of wound healing [124,125,126].

## 8. Concluding Remarks

The organization and composition of the cell membrane is a crucial factor during T cell activation. The initial signaling cascade, as well as the cytoskeletal and intracellular traffic, produce the quick polarization of organelles to the IS and the rapid shift in protein and lipid content. Mitochondria, peroxisomes, and the endolysosomal system are key players in lipid and protein homeostasis. In addition, they regulate both vesicle traffic and exosome production and their cargo content. Lipids act as mediators of signaling events, scaffolds for proteins in a structural and/or dynamic way, and as energy suppliers. When energy demands increase, the glycolytic pathway is promoted, causing an ATP increment associated to metabolic reprogramming. At this point, lipids are not further required as energy providers and become players in IS signaling. These molecules act as important modulators altering plasma membrane composition, endolysosomal biogenesis and dynamics, and regulating ROS levels. These roles place lipids as important regulators of the immune synapse and, in turn, of T cell activation. In fact, alterations in lipid homeostasis are related to several diseases of the immune system. Altogether, this information emphasizes the importance of understanding the normal and pathological functions of lipids in relation to T cell signaling.

## Figures and Tables

**Figure 1 ijms-21-02859-f001:**
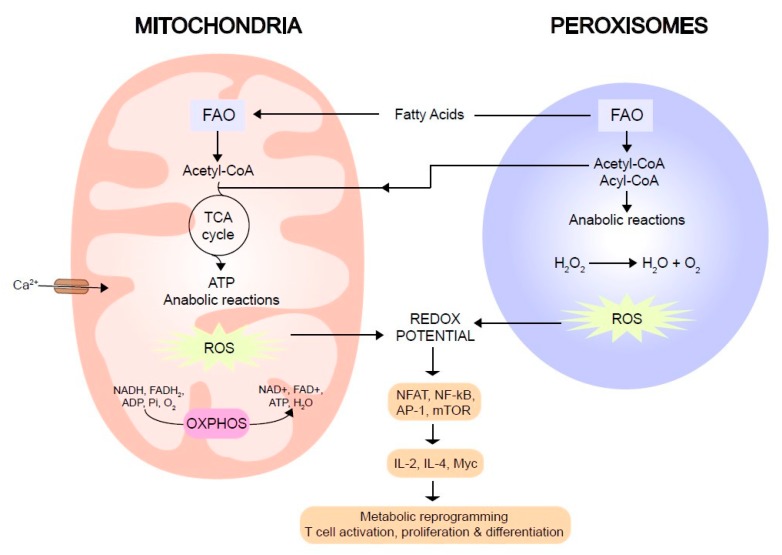
Reactive oxygen species (ROS) and lipids interplay between mitochondria and peroxisomes. Lipid metabolism is tightly regulated during T cell activation. Catabolic routes for energy production as fatty acid oxidation are located inside mitochondria and peroxisomes. The enzymatic production of ROS and different routes regulating their intracellular accumulation also localize at mitochondria and peroxisomes. These processes have an important impact on the transcription of genes, such as cytokines, involved in T cell differentiation and regulatory and effector responses. ROS: Reactive Oxygen Species.

**Figure 2 ijms-21-02859-f002:**
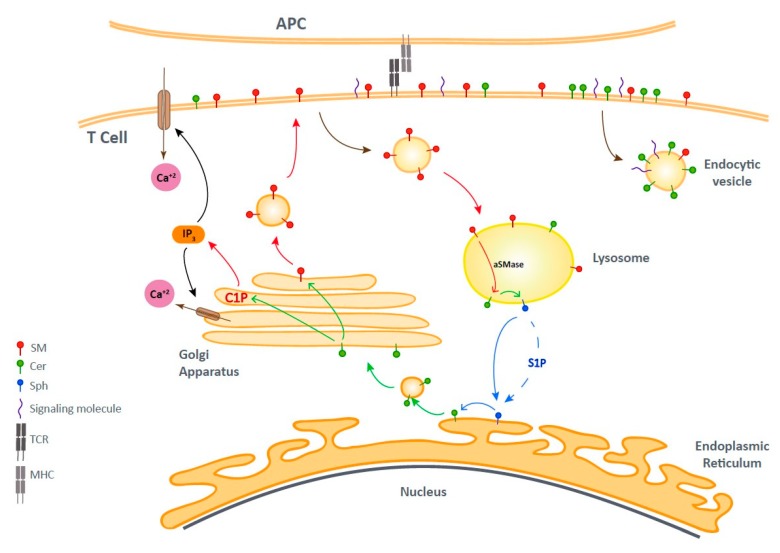
The cycle of bioactive sphingolipids upon T cell activation. T cell receptor (TCR) stimulation produces endocytic vesicles that regulate signaling. Sphingomyelin (SM) transport to lysosomes by endocytic vesicles allows its conversion to Cer and Sph. Then, the direct or mediated arrival of Sph to the endoplasmic reticulum (ER) leads to its re-conversion into Cer, which goes to the GA through vesicular transport. In the GA, Cer is phosphorylated and ceramide-1-phosphate (C1P) induces Inositol-1, 4, 5-trisphosphate (IP_3_) production, leading to an opening of plasma membrane and ER Calcium channels that regulates TCR signaling. Besides that, in the ER, Cer is converted to SM, which is then recycled to the plasma membrane, thereby ending the cycle. TCR: T Cell Receptor; Cer: Ceramide; Sph: Sphingosine; ER: Endoplasmic Reticulum; GA: Golgi Apparatus; C1P: Ceramide-1-Phosphate; IP_3_: Inositol-1, 4, 5-trisphosphate.

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
