# Peer review of "The Swing of Lipids at Peroxisomes and Endolysosomes in T Cell Activation"

_ijms, 2020, doi:10.3390/ijms21082859_

Round 1

Reviewer 1 Report

Dosil and colleagues presented a detailed review that summarizes the contributions of endolysosomes, lipids, mitochondria and peroxisomes during T cell activation. The review is well written and provides a quick source of information on mechanisms of immunological synapse.

Comments:

  1. Line 25: ‘citotoxicity’ should be replaced with ‘cytotoxicity’
  2. Line 28: ‘Immune Synapse’ should be ‘immune synapse’
  3. Lines 34, 35 & 57: Please write the full names for APC, TCR, MHC and ROS. The full names written in the abstract are not sufficient
  4. Line 88: Please write a line to explain Hrs
  5. Line 214: The authors mentioned some studies on mTORC1 without including multiples references to support the statement
  6. Line 304: ‘diseases as asthma’ should be ‘diseases like’ or ‘diseases such as’

Author Response

We thank the reviewer by completing the review of our manuscript and by his/her suggestions and comments.

1, 2, 3 and 6: Referee is right, and we have modified the text accordingly.

4: Referee´s point is important, and indeed this point was not properly documented. Thus, we have included the spelling of Hrs and described its main role at intracellular traffic.

5: There was a break in the paragraph and the studies referred where distant from the text. We have fixed this point.

Reviewer 2 Report

Overall the manuscript is excellently written and the topic its very interesting for immunologists.

Just some minors comments:

  1. Please name the IS as immunological synapse or immune synapse, just choose one to be consistent along the manuscript.
  2. Although the IS formed by T cells and APC are the most studied and canonical, there are others IS involving others immune cells, like NK cells, mast cells, etc. I suggest that this point must be clarify.
  3. Line 35: TCR receptor, change it by TCR or T cell receptor (TCR).
  4. Line 43: change T:APC by T cell:APC.
  5. Lines 89 and 246: change peripheral supramolecular clusters by peripheral supramolecular ACTIVATION clusters.
  6. Given that authors comment that (line 78)“The ELS uses MVB morphogenetic pathways to reach the lumen of lysosomes, where intracellular and extracellular cargoes are degraded. This process acts in concert with autophagy, in which damaged organelles and aggregated proteins are recycled through lysosomal pathways”. Just as suggestion, could the authors provide a brief comment about the literature in the role of autophagy in the regulation of IS signaling? I think this is a very relevant topic.
  7. Line 197: I would change “with the same pathogen” by “with the same antigen”
  8. Some references contain “(in eng)”, please delete it.

Author Response

We thank the reviewer by his/her criticism; we have addressed all the comments and suggestions and feel that the manuscript has clearly improved.

1, 2, 3, 4, 5, 7 and 8: we have changed the terms as suggested, and deleted the term (in eng), which was a typo error introduced by the endnote.

6: We agree with the reviewer in that this is a relevant issue that was omitted in the text, and we have included some recent evidence that highlights the relevance of the autophagy process and its connection with the immune synapse. Relevant bibliography has also been quoted.